SOFTWARE

# Accelerating joint species distribution modelling with Hmsc-HPC by GPU porting

**Anis Ur Rahman**[1], **Gleb Tikhonov**[2], **Jari Oksanen**[2], **Tuomas Rossi**[3], **Otso Ovaskainen**[1,2]*

**1** Department of Biological and Environmental Science, Faculty of Mathematics and Science, University of Jyväskylä, Jyväskylä, Finland, **2** Organismal and Evolutionary Biology Research Programme, Faculty of Biological and Environmental Sciences, University of Helsinki, Helsinki, Finland, **3** CSC – IT Center for Science Ltd., Espoo, Finland

* otso.t.ovaskainen@jyu.fi

**Data Availability Statement:** The Hmsc-HPC add-on package is available for download from its designated GitHub webpage https://github.com/hmsc-r/hmsc-hpc. The repository includes preprocessed Australian plant data used in this

## Abstract

Joint species distribution modelling (JSDM) is a widely used statistical method that analyzes combined patterns of all species in a community, linking empirical data to ecological theory and enhancing community-wide prediction tasks. However, fitting JSDMs to large datasets is often computationally demanding and time-consuming. Recent studies have introduced new statistical and machine learning techniques to provide more scalable fitting algorithms, but extending these to complex JSDM structures that account for spatial dependencies or multi-level sampling designs remains challenging. In this study, we aim to enhance JSDM scalability by leveraging high-performance computing (HPC) resources for an existing fitting method. Our work focuses on the `Hmsc` R-package, a widely used JSDM framework that supports the integration of various dataset types into a single comprehensive model. We developed a GPU-compatible implementation of its model-fitting algorithm using Python and the `TensorFlow` library. Despite these changes, our enhanced framework retains the original user interface of the `Hmsc` R-package. We evaluated the performance of the proposed implementation across various model configurations and dataset sizes. Our results show a significant increase in model fitting speed for most models compared to the baseline `Hmsc` R-package. For the largest datasets, we achieved speed-ups of over 1000 times, demonstrating the substantial potential of GPU porting for previously CPU-bound JSDM software. This advancement opens promising opportunities for better utilizing the rapidly accumulating new biodiversity data resources for inference and prediction.

## Introduction

The past decade has seen a transformative revolution in data acquisition and sampling methodologies, making large-scale data accessible for ecological research on species communities [1]. This emergence of novel data resources not only enhances our fundamental understanding of biodiversity but also establishes a new foundation for sustainable management in the context of global change. However, converting this data into reliable scientific insights presents

study as CSV tables. Moreover, the repository also includes the R codes used in the Australian plant datas.

**Funding:** This project was funded by the Academy of Finland (grant no. 336212 and 345110) to OO, and the European Union: the European Research Council (ERC) under the European Union's Horizon 2020 research and innovation programme (grant agreement No 856506 to OO; ERC-synergy project LIFEPLAN) and the HORIZON-INFRA-2021-TECH-01 project 101057437 (Biodiversity Digital Twin for Advanced Modelling, Simulation and Prediction Capabilities) to OO. Views and opinions expressed are those of the author(s) only and do not necessarily reflect those of the European Union or the European Commission. Neither the European Union nor the European Commission can be held responsible for them. The funders had no role in study design, data collection and analysis, decision to publish, or preparation of the manuscript.

**Competing interests:** The authors have declared that no competing interests exist.

significant challenges in terms of data processing and interpretation. A key statistical development in species community analysis is joint species distribution modelling (JSDM), which combines Species Distribution Modelling with ordination-like approaches to community assembly assessment [2, 3]. JSDM has become widely used by ecologists and has attracted significant interest from statisticians for further methodological development. Several user-friendly software packages have been created, offering diverse model structures and fitting algorithms. [4–11].

Since their introduction, the computational scalability of JSDMs for large community data has been a critical research focus. Initially formulated as a covariate-constrained multivariate probit (MVP) model [2], JSDMs faced challenges with the small-n-large-p statistical issue when applied to many species, due to the quadratic growth in estimated parameters with species count. This limitation prompted the search for alternative formulations, leading to the widespread adoption of generalized linear latent variable models (GLLVM) [3] as the most common design. Subsequent developments integrated Gaussian Processes to account for potential spatial or spatiotemporal autocorrelation in community data [12, 13], which initially scaled poorly with the number of spatial locations. Further progress in spatial statistics has since enhanced the scalability of these spatial JSDMs [14, 15]. Meanwhile, research has also significantly advanced the computational efficiency of classic GLLVM fitting through novel techniques for approximating the integration of latent variables [9, 16]. Additionally, scalability improvements for previously limited MVP-based JSDM formulations have included strategies such as Monte Carlo integration with stochastic gradient descent and mini-batching [10], as well as a two-stage divide-and-conquer approach [17].

Recent developments in model-fitting approaches have enhanced the computational scalability of GLLVM and MVP. However, these improvements have not been extended to more complex JSDM variants. Complex JSDMs not only link species data with environmental predictors but also explore dependencies on species traits or phylogenetics. Additionally, ecologists often employ hierarchical multi-level designs [18], which GLLVM or MVP only partially addresses by assuming the statistical interchangeability of observed sampling units after adjusting for covariates. As ecological datasets expand, their heterogeneity typically increases, underscoring the need for accelerated model-fitting methods, especially for large data studies. Motivated by this unmet need, our study explores a complementary approach to accelerating JSDM fitting: instead of developing new model-fitting algorithms, we aim to accelerate an existing algorithm through a novel implementation that leverages modern high-performance computing (HPC) resources.

In this work, we focus on the hierarchical modelling of species communities (HMSC) [7]. HMSC allows researchers to estimate how species occurrences depend on environmental predictors and how species-environment relationships are influenced by species traits and phylogenetic relationships. It further accommodates various study designs, including spatially or temporally explicit data and multi-level designs [19]. The HMSC framework is implemented in the `Hmsc` R-package, which enables users to define and fit various model types, and subsequently post-process model outputs for prediction and inference [20]. While the `Hmsc` R-package has demonstrated strong predictive performance [21], it faces computational inefficiencies with large datasets [22], prolonging model-fitting processes and limiting its utility for extensive ecological datasets, even with simpler GLLVM models [10].

To address this computational bottleneck, we developed `Hmsc-HPC` package, an extension that enhances the functionality of the `Hmsc` R-package by efficiently utilizing graphical processing units (GPUs) and HPC infrastructure. The central innovation of `Hmsc-HPC` is the acceleration of the model fitting process by transitioning from R-coded calculations to a TensorFlow-based computational backend capable of leveraging GPU acceleration [23]. By

harnessing the parallel processing capabilities of GPUs, we significantly speed up the execution of the block-Gibbs sampler used in HMSC fitting. This acceleration markedly reduces the time required for model fitting and addresses performance limitations related to dataset size, thereby improving the package's capability to handle large datasets that were previously computationally prohibitive.

## Design and implementation

### Integrating `Hmsc-HPC` with `Hmsc` R-package

The `Hmsc` R-package provides a comprehensive statistical framework for integrating data on species occurrence records, environmental covariates, species traits, and phylogenetic information while accounting explicitly for hierarchical, spatial, or temporal study designs [19]. Statistically, HMSC combines elements of generalized linear models and latent factor approaches within a hierarchical modelling framework. HMSC is fitted using Bayesian inference via Markov chain Monte Carlo (MCMC) sampling. The `Hmsc` R-package enables ecologists to define models, fit them to data, and utilize the fitted models for prediction and inference (Fig 1):

1. **Setting model structure and data:** In the first step, the user defines the desired model structure, which reflects the specifics of the collected data and is tailored to address specific ecological questions of interest. This step also includes pre-processing the original data into the assumed input format, specifying the optional model features and fine-tuning prior distributions of the model if necessary.

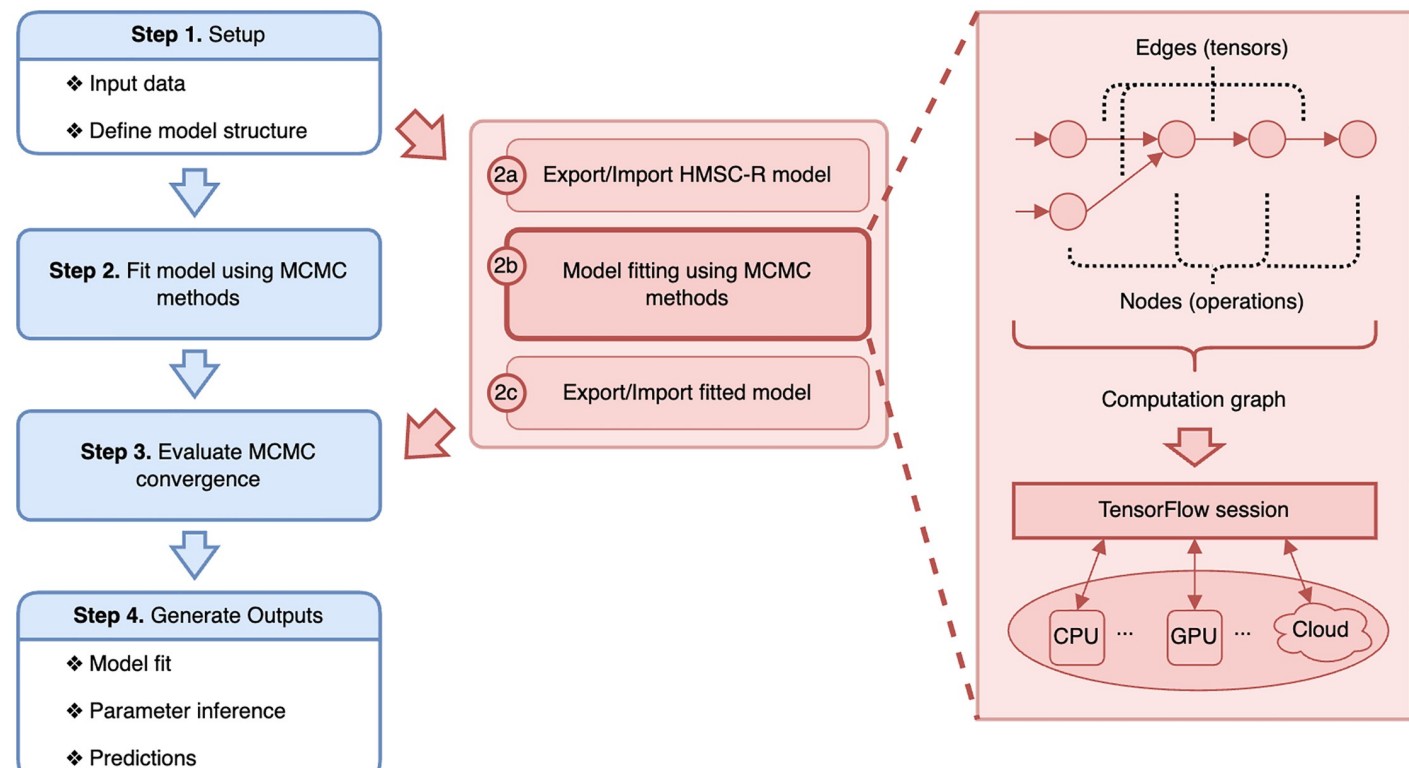

**Fig 1. Pipeline of community analysis with HMSC.** The left path represents the standard approach using the `Hmsc` R-package [20]. The alternative on the right describes the newly developed `Hmsc-HPC` augmentation, which allows for model-fitting deployment on HPC infrastructure.

2. **Model fitting:** The defined model is then fitted to the provided data using MCMC sampling. Specifically, the `Hmsc` R-package employs a block-Gibbs sampler, which iterates over blocks (naturally defined subsets) of HMSC parameters. Within each block, parameters are updated collectively while keeping others fixed at their current values, sampled from analytically derived conditional distributions due to the conditionally conjugate design of HMSC. These distributions are available in closed form. Subsequently, the resulting MCMC chains undergo processing through trimming and thinning to mitigate the effects of initial chain positions and autocorrelation. This refinement aims to approximate the true, non-tractable posterior parameter distribution in HMSC using a finite number of samples.

3. **MCMC fit diagnostics:** In this step, the validity of the acquired MCMC chains is assessed using both formal criteria (such as Gelman-Rubin diagnostics [24]) and visual inspection of trace plots.

4. **Inference and predictions:** After obtaining a reliable posterior approximation, this step encompasses the operations that the user wishes to perform with the fitted model. These operations may include evaluating the model's explanatory and predictive capabilities, exploring parameter estimates, and making predictions [7, 19].

The practical use of the Hmsc R-package for large models is hindered by the computational intensity of Step 2, due to two main reasons. Firstly, Bayesian inference with MCMC methods is inherently more computationally demanding than approximate Bayesian methods, such as integrated nested Laplace approximations (INLA) or empirical Bayes methods. These approximate methods are designed to be more computationally efficient while still providing reliable estimates, unlike MCMC which aims to assess the entire posterior distribution–a significantly more complex and resource-intensive task. Secondly, from a software engineering standpoint, Step 2 in the Hmsc R-package primarily relies on native R computational routines, which are less numerically efficient compared to implementations in compiled programming languages [25]. This reliance on a less efficient computational framework results in significantly poorer performance for the Hmsc R-package compared to recently introduced alternatives tailored for highly accelerated model fitting [10].

Here we overcome the computational limitations of the Hmsc R-package by replacing its original model-fitting implementation with a version designed for HPC hardware. This new package, called Hmsc-HPC, is optimized for deployment on GPUs, which have demonstrated their effectiveness in various computationally intensive tasks beyond their traditional graphical applications in recent years. Despite the typically sequential nature of MCMC fitting algorithms, the computations within each MCMC step can benefit significantly from breaking down into small, independent tasks that can run concurrently on GPU cores. The majority of operations within the block-Gibbs algorithm are algebraic in nature, lending themselves well to a "single instruction, multiple data" paradigm. By parallelizing these computations across the processing units of a GPU, calculations can be executed simultaneously, leading to substantial efficiency gains [26]. Furthermore, the implementation can run on multicore CPUs as well as GPUs, leveraging Python and TensorFlow dependencies for optimal performance across different devices.

To implement Hmsc-HPC, we redesigned and re-implemented the block-Gibbs sampler routine of the Hmsc R-package using Python language and the TensorFlow library [23]. We opted for TensorFlow over lower-level GPU programming options due to its robust application programming interface, which includes a wide array of linear algebra and statistical routines. This choice significantly reduced the need for developing standard low-level numerical

operations or relying on third-party tools. TensorFlow's computational graph concept is fundamental to its functionality and efficiency. It represents the entire computation algorithm as a directed graph, where nodes correspond to mathematical operations and edges denote the flow of data between these operations. This graph-based approach offers several advantages, such as portability, optimization opportunities, and support for distributed computing. Crucially, it decouples the definition of computation from its execution, enabling lazy evaluation or deferred execution. This separation allows TensorFlow to optimize the computation graph by combining operations for efficiency and potentially distributing non-sequential computations across multiple devices for concurrent processing. Graph execution involves running computations on the constructed graph, typically within a session context. This process entails evaluating specific nodes or operations to generate results. The clear distinction between graph construction and execution not only enhances TensorFlow's performance but also facilitates flexibility and extensibility in constructing complex data processing pipelines [27].

In addition to developing the Python code with the TensorFlow-based block-Gibbs sampler implementation, we made corresponding modifications to the Hmsc R-package. These changes facilitate the seamless integration of HPC resources during the model fitting phase, as illustrated in Fig 1:

1. **Setting model structure and data:** Initially, the user defines the model within R using the standard workflow of the Hmsc R-package. This step also involves initializing the starting positions of the MCMC chains.

2. **Exporting a specified model to Python:** At this stage, the user saves the model as an RDS file within the R session, which includes the model specification and MCMC settings.

3. **Model fitting with TensorFlow:** The core step involves reading the RDS file into an independent Python session. Model fitting occurs in Python using the GPU-compatible MCMC sampler implemented in TensorFlow. This process begins with a compilation of the TensorFlow computational graph, which is subsequently executed. The TensorFlow graph can be executed on any device with appropriate Python and TensorFlow configurations. For instance, it can run on a CPU in the user's laptop and still offer significant speed improvements in model fitting. However, our implementation is optimized for GPUs dedicated to scientific computations, aiming to achieve the maximum performance for computationally intensive tasks.

4. **Exporting the posterior to R:** Once the posterior distribution is obtained, the user saves it as an RDS file within the Python session.

5. **Diagnostics, inference, visualization, and prediction:** In this final step, the user imports the RDS file back into an R session. Post-fitting procedures continue within the standard workflow of the Hmsc R-package, independent of the framework used for the model fitting.

In addition to leveraging intra-chain low-level parallelization to harness GPU cores for enhanced performance, we have also implemented an interface that allows for the simultaneous computation of multiple MCMC chains. This approach is particularly advantageous in HPC infrastructures, such as a computational cluster, where multiple jobs from a user can be placed simultaneously. The internal job scheduler then assigns these jobs to available computational nodes automatically. Since MCMC chains operate independently and do not require communication with each other, this setup is ideal for parallelization across chains. Each job can handle the computation of one or a few chains, maximizing efficiency and utilizing computational resources effectively.

In summary, our approach retains the user interface of the Hmsc R-package while enabling the utilization of HPC infrastructure and GPU acceleration during the most computationally intensive step. This integrated, cross-language workflow leverages the strengths of R, Python, and TensorFlow to provide a streamlined and efficient methodology for analyzing large species community data with models that can account for complex dependency structures.

## Performance comparison

We conducted a performance assessment of the GPU-accelerated HMSC implementation across different models characterized by varying data dimensions, including the numbers of sampling units and species analyzed. Additionally, we explored diverse model structures that varied in whether or not phylogenetic information was included, the design of random levels, and the choice of spatial method. The execution of the Hmsc R-package used as a baseline reference, was performed and timed on a consumer-level laptop CPU. In contrast, the Hmsc-HPC implementation was run on NVIDIA Volta V100 GPUs located in the AI partition of the Puhti cluster, operated by CSC—IT Center for Science, Finland.

We set up the computational performance experiment using the same case study as previously employed to compare the numerical performances of alternative spatial methods in the Hmsc R-package implementation [14]. This case study serves as an illustrative example of the challenges encountered when using the Hmsc R-package with large datasets, even for relatively simple models. The dataset consists of spatially-referenced recordings of plant presence-absence in South-West Australia, along with several environmental predictors. After excluding most rare species (those with fewer than five occurrences), the dataset comprises of 622 species recorded at 25,955 locations. Additionally, the dataset includes a set of binary traits for each species. We also incorporated information regarding species' taxonomy based on the GBIF Backbone taxonomy [28].

We conducted experiments closely following the design of the original study [14], evaluating the model fitting performance for five distinct variants of HMSC models. These model variants cover different approaches to handle the spatial context of the data [12, 14]: 1) a spatially igno-rant model, 2) a spatial model with a full Gaussian process (GP) structure for latent factors, 3) a spatial model with predictive GP approximation [29] (PGP, with 55 predictive process knots distributed along a uniform hexagonal grid spanning the study area), 4) a spatial model with near-est neighbour GP approximation [30] (NNGP, with ten neighbours), and 5) a spatially ignorant model that accounts for taxonomical relationships, thus using taxonomy as a proxy for phylogeny to estimate the phylogenetic signal in species responses to environmental predictors [7]. All models included the data on species traits. We introduced variation in two key dimensions: the number of species ($n_s$ = {40, 160, 622}) and the number of sites ($n_y$ = {100, 200, 400, 800, 1600, 3200, 6400, 12800, 25955}). This allowed us to create diverse sub-datasets, ranging from reason-ably small data sizes to fairly large datasets. To facilitate comparison across varying data dimen-sions, we kept the number of latent factors fixed at 10. For each sub-dataset, we fitted each of the five HMSC variants with both the Hmsc R-package and Hmsc-HPC, excluding cases for which the full GP spatial model was infeasible due to insufficient RAM/VRAM. We recorded the total fitting time for each model-dataset combination and quantified the execution time required for a single cycle of the block-Gibbs sampler. While different models may require different numbers of samples to converge, our primary focus is the relative comparison between alternative imple-mentations of the same mathematical algorithm. Therefore, comparing the computational effort per Gibbs cycle provides a relevant characterization of the observed differences.

## Results

The results presented in Fig 2 clearly demonstrate the computational benefits of fitting HMSC models on a high-end specialized GPU device. The speed-up of Hmsc-HPC over the Hmsc R-

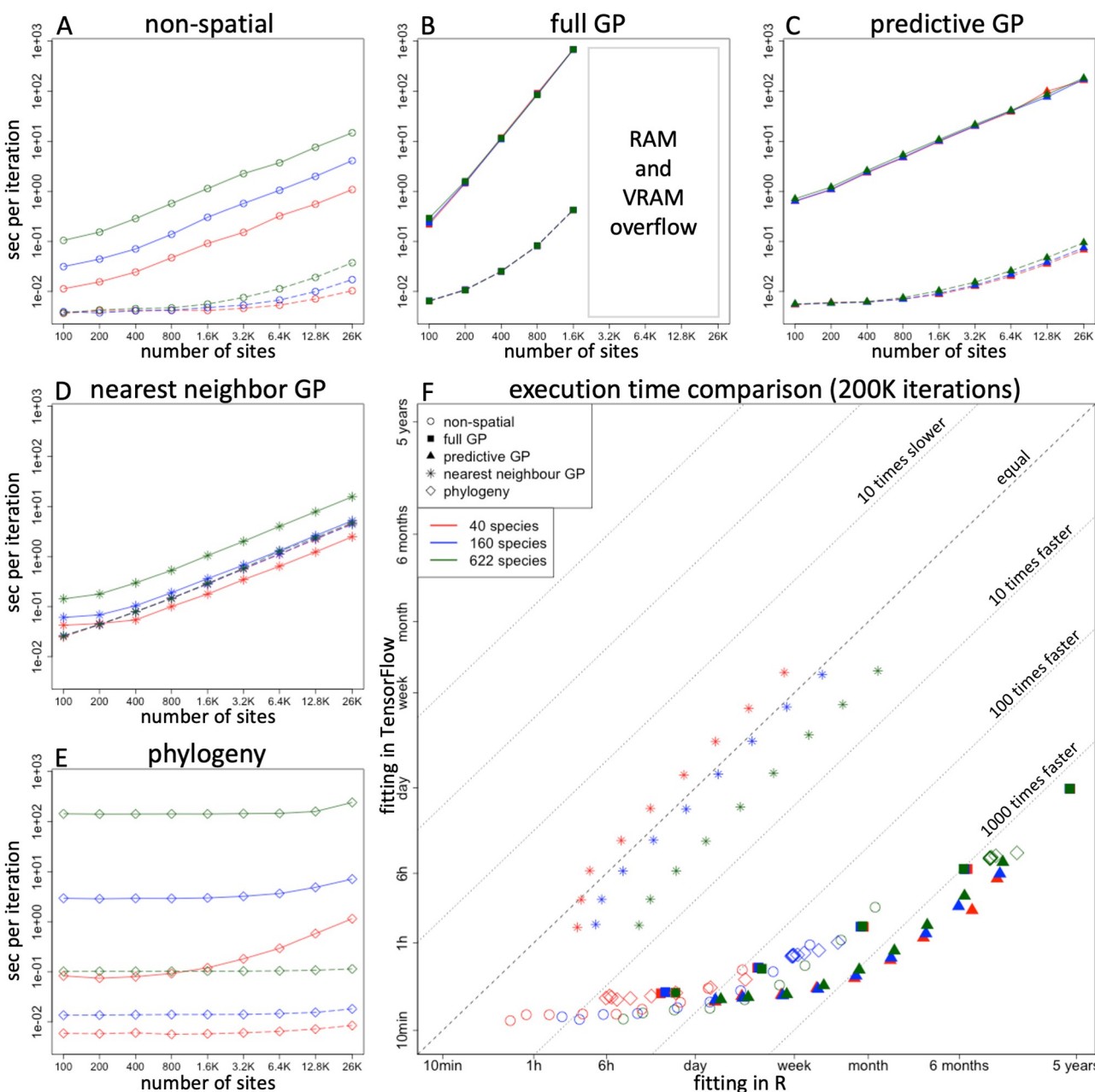

**Fig 2. Performance comparison on GPU.** Panels A-E represent execution time per block-Gibbs cycle for different HMSC models with respect to the number of sites and species. Red, blue and green colours stand for $n_s$ = 40, 160 and 622 species correspondingly. Solid lines depict the execution times for model fitting with the R backend, and dashed lines for fitting with the TensorFlow backend. In some panels, the lines of different colours overlap due to the minor effect of the number of species on execution times. Panel F summarizes the overall comparison of execution times between the R backend and the TensorFlow backend. This figure shows the same data as depicted in panels A-E, but the times are multiplied by 200,000 to exemplify total computation times for a typical number of MCMC iterations in a model fit with a transient of 100,000 and 1000 samples obtained with a thinning of 100. Dashed grey lines indicate ratios between TensorFlow and R computational performance, thus showing how many times faster or slower TensorFlow is compared to R. The symbols and colours represent different model types and numbers of species, matching those used in Panels A-E (as indicated in the legend).

package increases with the size of the dataset and the overall computation time. Notably, even for a non-spatial model, the GPU-based execution outperforms the similar R-based execution by nearly 1000 times for the largest dataset, equating to an acceleration of three orders of magnitude. For full GP and PGP models, this acceleration is evident even with smaller numbers of sites, as these models involve operations with large matrices that are highly suitable for GPU acceleration. Models incorporating taxonomy exhibit a similar pattern, but the acceleration is more closely aligned with the number of species rather than the variation in the number of sites.

In contrast to the other model variants, the results for the NNGP models did not indicate a clear benefit from using Hmsc-HPC. The HMSC algorithm for handling the NNGP approximation for spatial latent factors relies on sparse linear algebra operations, specifically the Cholesky decomposition of sparse symmetric positive definite matrices and left-hand division with sparse triangular matrices. Currently, these operations are not available in TensorFlow. To address this, we used the capability to "inject" the necessary sparse operations from another Python package into the TensorFlow graph, resulting in these operations being conducted on the CPU. The introduced device-host communication overhead appears to bottleneck the overall execution, diminishing the benefits of accelerated GPU execution for the remaining operations. The NNGP approaches have been identified as superior to low-rank representations of spatial covariance matrices [14, 30, 31], and thus solving this bottleneck is a key challenge for future work. However, we note that the current Hmsc-HPC package enables fitting models to large spatial data using the GP and especially the PGP approaches.

## Conclusion

In this study, we introduced a novel `Hmsc-HPC` package that provides a parallel and efficient implementation of HMSC fitting using a GPU-compatible TensorFlow backend. This development is designed as an optional add-on to the existing, well-established `Hmsc` R-package to enhance its usability for JSDM practitioners. Our evaluation, conducted on an extensive dataset of species occurrence records, demonstrates that the new implementation yields speed improvements of over 1000 times compared to the `Hmsc` R-package approach when the model fitting problem is sufficiently computationally intense to start with (Fig 2). This effectively means that models which previously required five years to fit can now be processed in just one day. The achieved speed-up significantly expands the practical boundaries of fitting HMSC models to large ecological datasets.

While GPU utilization is standard in machine learning, its application in statistical modelling remains relatively uncommon. To our knowledge, among existing JSDM software, only `s-jSDM` [10] is designed to leverage GPU devices. Our findings on the speed-up achieved through GPU porting complement recent developments in model fitting approaches that have greatly advanced scalability for JSDMs, particularly on structurally simpler models. These results underscore the potential benefits of further developing JSDMs to target GPU acceleration. However, our work also highlights challenges with GPU porting, such as the limited availability of efficient sparse linear algebra routines in TensorFlow, which hindered the implementation of computationally efficient approaches for NNGP models.

Looking forward, we outline four directions for advancing Hmsc-HPC and potentially other GPU porting efforts in JSDM. First, recent studies have extended the NNGP method [31], previously viewed as the pinnacle of scalable spatial statistics, to generalize neighborhood-only conditional dependence to blocks of spatial locations [32] and proposed algorithms independent of sparse linear algebra [33]. GPU-based implementations of such block spatial models may be computationally advantageous over the NNGP models. Second, while here we implemented a fixed floating-point precision (FPP), it may be advantageous to use mixed FPP

within the model fitting algorithm [34], thereby leveraging enhanced GPU acceleration with low FPP for precision-insensitive calculations. Third, leveraging TensorFlow's auto-differentiation capabilities enables efficient integration of gradient-based generic MCMC samplers like Hamiltonian Monte Carlo [35] into block-Gibbs fitting algorithms used in Hmsc-HPC. This hybrid MCMC sampling approach may mitigate the drawbacks of individual sampling schemes, yielding universally superior MCMC convergence properties at the cost of minor additional computational overhead [36]. Finally, we aim to broaden our evaluation across diverse datasets and hardware configurations, including those incorporating explicit phylogenetic information. These efforts will enhance our understanding of the computational and predictive capabilities of our enhanced HMSC framework.

## Acknowledgments

We thank CSC—IT Center for Science, Finland for providing access to HPC infrastructure and high-end GPU devices. Jesse Harrison and Jussi Heikonen from CSC significantly contributed to the design and support of this work on CSC computing platforms. Matt White kindly provided a taxonomy of the plant species used in the case study. We appreciate discussions and initial explorations with Graham Taylor and Sara El-Shawa on alternative HPC software platforms, among which we selected TensorFlow for this work.

## Author Contributions

**Conceptualization:** Anis Ur Rahman, Gleb Tikhonov, Jari Oksanen, Otso Ovaskainen.

**Data curation:** Gleb Tikhonov, Otso Ovaskainen.

**Formal analysis:** Gleb Tikhonov.

**Funding acquisition:** Otso Ovaskainen.

**Investigation:** Anis Ur Rahman, Gleb Tikhonov, Jari Oksanen, Otso Ovaskainen.

**Methodology:** Anis Ur Rahman, Gleb Tikhonov, Jari Oksanen, Tuomas Rossi.

**Project administration:** Gleb Tikhonov, Otso Ovaskainen.

**Resources:** Anis Ur Rahman, Gleb Tikhonov, Jari Oksanen, Tuomas Rossi, Otso Ovaskainen.

**Software:** Anis Ur Rahman, Gleb Tikhonov, Jari Oksanen, Tuomas Rossi.

**Supervision:** Jari Oksanen, Otso Ovaskainen.

**Validation:** Anis Ur Rahman, Gleb Tikhonov, Jari Oksanen, Tuomas Rossi, Otso Ovaskainen.

**Visualization:** Anis Ur Rahman, Gleb Tikhonov.

**Writing – original draft:** Anis Ur Rahman, Gleb Tikhonov.

**Writing – review & editing:** Anis Ur Rahman, Gleb Tikhonov, Tuomas Rossi, Otso Ovaskainen.

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
