## [Decision Letter · Decision Letter 0]

16 Apr 2024

Dear Dr Rahman,

Thank you very much for submitting your manuscript "Accelerating joint species distribution modeling with Hmsc-HPC: A 1000x faster GPU deployment" for consideration at PLOS Computational Biology.

As with all papers reviewed by the journal, your manuscript was reviewed by members of the editorial board and by several independent reviewers. In light of the reviews (below this email), we would like to invite the resubmission of a significantly-revised version that takes into account the reviewers' comments.

I agree with the reviewers that this is manuscript presents an interesting approach to a persistent problem in ecology. However, I also agree that more work is needed to appropriately describe the direct contributions of this work and to ground the work in existing literature. The authors should also consider modifying the title to more accurately describe what the 1000x speed-up is relative to and/or simply remove that from the title. I do also agree with reviewer 3 that fewer end users may be familiar with Python and it is worth a few sentences outlining the added complexity of both running code against GPUs and using Python as compared to R (I'm not trying to start an argument about the merits of various languages, but I doubt many people would defend Python's package mgmt).

We cannot make any decision about publication until we have seen the revised manuscript and your response to the reviewers' comments. Your revised manuscript is also likely to be sent to reviewers for further evaluation.

Sincerely,

Samuel V. Scarpino

Academic Editor

PLOS Computational Biology

James O'Dwyer

Section Editor

PLOS Computational Biology

I agree with the reviewers that this is manuscript presents an interesting approach to a persistent problem in ecology. However, I also agree that more work is needed to appropriately describe the direct contributions of this work and to ground the work in existing literature. The authors should also consider modifying the title to more accurately describe what the 1000x speed-up is relative to and/or simply remove that from the title. I do also agree with reviewer 3 that fewer end users may be familiar with Python and it is worth a few sentences outlining the added complexity of both running code against GPUs and using Python as compared to R (I'm not trying to start an argument about the merits of various languages, but I doubt many people would defend Python's package mgmt).

Reviewer's Responses to Questions

**Comments to the Authors:**

Reviewer #1: Review of "Accelerating joint species distribution modeling with Hmsc-HPC: A 1000x faster GPU deployment

Rahman et al. present an exciting GPU implementation of Hmsc to fit a multitude of complex joint species distribution models. This is a very impressive contribution, and one that will be utilized greatly by the ecological community. Overall, I found the manuscript an appropriate length for communicating a software advancement. However, I found certain aspects of the manuscript to potentially overstate the value of Hmsc-HPC, particularly in the context of fitting spatially-explicit models, which I believe the authors should address prior to publication.

Major comments

1. Benefits of Hmsc-HPC for spatial models: Figure 2 clearly highlights the benefits of Hmsc-HPC for non-spatial and phylogenetic models, which is quite exciting given these models likely comprise a large majority of the models fit by the Hmsc user community. However, as the authors point out, there were minimal benefits, if any at all, for the GPU implementation of NNGP models, which also showed fairly close correspondence with the model run times for the GP and predictive process models when implemented in Hmsc-HPC. This leads me to ask the question: is it worthwhile for an Hmsc user who wants to fit a large spatial model to use the GPU implementation over the existing Hmsc-R package functionality for NNGPs? It is clear from Panel B that Hmsc-HPC yields clear benefits for fitting full GP models, but given the very good performance of NNGP models at closely approximating a full GP (e.g., Datta et al. 2016), is it realistic to expect a GP JSDM to be fit by an ecologist with anything more than 100-200 spatial locations? Further, while the predictive process model also showed substantial improvements in Hmsc-HPC, predictive process models and other low rank approximations to GPs are known to oversmooth (Stein 2014), which is something many of the authors of this manuscript found in a previous manuscript with this exact data set (Tikhonov et al. 2020). NNGPs do not have the same oversmoothing issues. While the predictive process models with Hmsc-HPC did have lower run times across the board compared to the NNGP models in Hmsc-R, it is not possible to directly compare the approaches due to (1) potential differences in mixing/convergence of the MCMC chains; (2) differences in model performance that may result from using too few knots in the predictive process approximation (here the authors use 55 while in Tikhonov et al. 2020 1026 knots were used, a substantial increase in run time) or too few neighbors (10 neighbors here vs. 20 in Tikhonov et al. 2020). I know this is not the objective of the performance comparison, as the authors point out in lines 210-214. However, given the lack of substantial improvements for the NNGP models, and the potential for the NNGP models implemented in Hmsc-R to outperform predictive process models implemented in Hmsc-HPC, I do not feel that there is conclusive evidence of Hmsc-HPC providing improvements in fitting spatial models relative to Hmsc-R. In my opinion, this would require a substantial comparison study of the models in terms of run times to convergence, predictive performance, etc. This would be a substantial undertaking, and something I do not think is necessary for this manuscript. I am not denying the benefits of Hmsc-HPC for fitting nonspatial or phylogenetic models; those findings are extremely clear. However, I believe the authors should more explicitly state that a more robust assessment/comparison of the NNGP/predictive process models under both platforms should be a future avenue of research in order to best determine the most effective approach to fitting spatial models.

2. Title: the "1000x faster GPU deployment" is context dependent, as it depends on the model type and size of the data set. The 1000x improvement, as the authors point out in the abstract, is for the largest data set (over 20,000 points), which I would guess is larger than the data sets most Hmsc users work with. Thus, I find this title overstated, and would suggest removing the "1000x faster GPU deployment" and instead only including that in the abstract where the proper context for it can be provided.

Minor comments:

+ Line 26-28: it may not be clear to ecologists reading the paper that "Hmsc-HPC" is not a new R package, and rather is an extension written in Python that needs to be installed separately from the Hmsc-R CRAN implementation. I would suggest including this information explicitly here.

+ Line 137-138: does this mean the user must have familiarity with Python, or is this all accomplished directly through R? Some more detail, and potentially a reference to an online example of how this is done with the package, would be helpful.

+ Lines 192-195: the authors do not give appropriate citation for the predictive process (Banerjee et al. 2008) or NNGPs (Datta et al. 2016). These should be cited accordingly.

References

+ Banerjee, S., Gelfand, A. E., Finley, A. O., & Sang, H. (2008). Gaussian predictive process models for large spatial data sets. Journal of the Royal Statistical Society Series B: Statistical Methodology, 70(4), 825-848.

+ Datta, A., Banerjee, S., Finley, A. O., & Gelfand, A. E. (2016). Hierarchical nearest-neighbor Gaussian process models for large geostatistical datasets. Journal of the American Statistical Association, 111(514), 800-812.

+ Stein, M. L. (2014). Limitations on low rank approximations for covariance matrices of spatial data. Spatial Statistics, 8, 1-19.

+ Tikhonov, G., Duan, L., Abrego, N., Newell, G., White, M., Dunson, D., & Ovaskainen, O. (2020). Computationally efficient joint species distribution modeling of big spatial data. Ecology, 101(2), e02929.

Reviewer #2: I revised the manuscript entitled “Accelerating joint species distribution modeling with Hmsc-HPC: A 1000x faster GPU deployment”. In this manuscript, authors introduced a new package to create JSDM. The manuscript is straightforward and clear, and the experiment performed to test the performance between Hmsc-R and Hmsc-HPC is appropriate. The tutorial provided also worked well and was very didactic. Perhaps, in the future, the package's authors could create a web page (e.g., with pkgdown) with tutorials for different modeling situations. One of the points I would like to highlight about the authors is that they integrate the two programming languages (R and Python). This will undoubtedly stimulate the use of Hmsc-HPC by former Hmsc-R users and others who only use R as a programming language. It is very exciting to see the speed at which JSDMs can be constructed, which in many other conditions would be prohibitive. I believe that the improvements that this package will bring to the already widely used Hmsc-R package will be of extreme importance for the development of these models, enhancing their use in new applications in regions with many samples or with many species.

Reviewer #3: The manuscript presents a new GPU-based, implementation of the HMSC fitting software for Joint Species Distribution Models. HMSC fits JSDMs in a Bayesian setting using MCMC, but since the original package implements those routines in the R programming language a large speed-up is to be gained by using the TensorFlow library in python for model fitting instead. This potentially greatly facilitates fitting JSDMs with complex model structures to large datasets in the Hmsc software.

Although the manuscript is generally well written, it contains some typos that I have attempted to outline in the detailed comments, but I generally recommend the authors to thoroughly re-read their text for potential errors that I (or another reviewer) might have missed. Generally, I find many sentences that are much less succinct than possible, and combined with the excessive use of unnecessary adjectives, the text at times feels convoluted and lacks the flow that makes for an enjoyable read.

I applaud and thank the authors for including a reproducible script of their analyses, and installation instructions for their new software package, this made trying out their implementation easier, generally more enjoyable, and trying the implementation clarified various aspects of the manuscript for me.

The selected literature in the first paragraphs is somewhat narrow, and although the article makes some mention of alternative (fast) software for JSDM fitting, it ignores large parts of the literature that were developed in recent years for fitting JSDMs. This includes the glmmTMB R-package, or the gllvm R-package whose main developer is of course at the same university as the last author of the current manuscript, but also e.g., copula-based methods (Popovic et all. (2022)), quasi-likelihod based methods (Kidzinski et al. (2022)), or the basis-function approach from Hui et al. (2023). Either how, Pichler and Hartig (2021) as a reference on L25 seems inappropriate as that reference does not speak to the “hinders of scientific discovery” due to computational limitations, but suggests a different method for fitting JSDMs based on an elastic net penalty.

There are various general statements on the benefit that this computational improvement is supposed to have on biodiversity research, e.g., “far-reaching implications for informing conservation strategies” or “a profound advancement in the domain of community ecology”. Although those might be true, since there is no comparison with other fitting methods (e.g., with the software by Pichler and Hartig (2021) that the authors cite), the reader is left wondering if this present GPU implementation is only a speed-up over the existing Hmsc software, or truly an advance for fitting JSDMs in all of community ecology as the manuscript states. Without comparison with other software implementations, it is not possible to assess the latter, and the manuscript feels like it is overselling its novelty, as it pertains a quite limited internal comparison based on a single dataset. My suggestion is that the authors tone down the language, or alternatively include a comparison with other software packages. The chosen title has a similar issue, since in the manuscript it becomes clear that there is not always a 1000 times speed-up, but only compared to the regular Hmsc implementation, and only for the single dataset with which a computational comparison has been performed.

Generally, I would appreciate some disclaimer in the text when it comes to user friendliness, as most ecologists fitting JSDMs will not be familiar with python (though it seems only a limited degree is necessary for use of HMSC-HPC), and performing a python installation, or debugging it, can be very challenging (and the vignette provided by the authors essentially states to google a problem if a user cannot figure it out, offering only very limited support even to the reviewers..fortunately I did not need any!). Out of curiosity, why did the authors decide to develop an add-on package, while keeping R as the default backend for HMSC, if use of the TensorFlow library has so much benefits?

Finally, I find the comparison based on a single dataset, and without any other software packages than that of the authors, very limiting. The authors even use a proxy for fitting Phylogenetic models as the dataset that they perform comparisons on does not actually seem to include Phylogenetic information. I advice the authors to expand on their comparison, use one or two more datasets, or just perform a complete set of simulations based on (a select few of) the various model structures that the Hmsc package is capable of fitting.

Detailed comments

Please number the manuscript throughout. The abstract and author summary were left unnumbered, leaving reviewers to manually count lines to provide comments, which is clearly not impossible, but unnecessary tedious.

Abstract

L3: “However, JSDM...” → “However, fitting or estimating JSDMs” or similar. Also, clarify what is meant here by “prohibitive”. I suppose you mean that the long computation time is prohibitive, but JSDMs in and of itself are not.

L6: Please rephrase. Your GPU implementation does not address computational limits of JSDMs generally, purely as implemented in the Hmsc software.

L9: “TensorFlow” → “the TensorFlow”

L10: “primarily targets to enable” → “enables”

L12: Please remove repeated statement from L10

L17: “boosts the scalability of Hmsc-R package” → “boosts scalability of the Hmsc R-package” and remove “reaching a significantly higher level” in the second part of the sentence

L19-L20: I will get back in my major comments about statements like these throughout the manuscript, but generally I think this is an overstatement w.r.t. the benefit of the improved method. It is great that models in Hmsc can now be fitted much faster, but there are other (potentially) faster algorithms that fit similar/the same JSDMs available, so to state that this particular GPU implementation is going to enable better informed conservation strategies, environmental management, and climate change adaptation planning, seems very unlikely.

Author summary

I somewhat take issue with the statement that the authors are “retaining the user-friendly R interface”, while later on L137-147 it becomes clear that the user actually needs to perform operations in an independent python session (albeit from R). I would like the authors to be more transparent on how their implementation requires interfacing independent use of R and python instead, as that is not a minor feat for many ecologists.

Introduction

L8: JSDMs were introduced >10 years ago, I am not sure I agree with the authors that I would call this “recent”

L13: The reference (4) cited in this sentence includes a comparison of predictive performance for JSDM fitting softwares and has little to do with fitting algorithms, which is what that this sentence is currently about. Please either 1) cite software packages or different fitting algorithms here instead, or 2) alter the sentence to target predictive performance as that is what this reference studies. Additionally, the reference is five years old and there have been many computational developments in fitting JSDMs that it does not cover.

L14: Difficult to read sentence, suggest rewriting to “HMSC facilitates researchers to study the environmental niche of species, potentially structured by additional traits and phylogenetic information” or similar.

L16: remove “regression-style” as it does not add anything to the sentence

L19: Again, I suggest changing “allows” to “facilitates” since researchers were also allowed to fit JSDMs before HMSC was developed.

L20: “post-process for prediction and inference” rarely can model results be processed for prediction and inference, since models are (should) usually be calibrated for one or the other (I have yet to meet a model that is suitable for prediction and inference). So I suggest the authors change this statement to “prediction or inference”.

L20: “Despite its utility” – please elaborate, what utility, or utility of what tool, do the authors refer here to?

L23-24: the models in HMSC can be fitted with different software, so I doubt that this statement is true. The availability of HMSC makes that these models can be fitted in a more user friendly setting, and by researchers with less knowledge of model fitting algorithms, so that it makes for generally more robust inference.

L29: “Optimizing” implies that the existing model fitting process is made more efficient, which I am not sure is the case here, because the GPU implementation actually facilitates the use of a different set of computational resources that were not previously available to the Hmsc-R user.

L37-L42: This statement may be true, although to be honest I doubt that it is. The authors do not provide an exhaustive comparison with other JSDM fitting softwares, so there is no way to verify that the software that the authors have developed actually fits these models faster than any other software currently available. And that would be a requirement for “a profound advancement in the domain of community ecology…”.

L47: “intricate” is probably not how most ecologists would describe their study designs. Perhaps “nested” or “complex” is more appropriate here.

L49: “Generalised regression” is not a method I am familiar with. Do the authors refer here to the GLM-framework?

L50: “resulted” → “resulting”

L51 “in Bayesian paradigm” → “in the Bayesian paradigm”

L52: MCMC does not make for a more versatile modeling framework than via other fitting algorithms, so please rephrase this statement.

L55: “Prediction and inference” → “Prediction or inference”, see comment w.r.t L20 where this was also stated in the same manner.

Fig 1. … “with Hmsc-R package” → “with the Hmsc-R package”. Also, throughout, please rephrase “Hmsc-R package” to “Hmsc R-package”.

Unfortunately, I could only access the figures in very low resolution as attached to the manuscript, so that they were near impossible to assess.

L75: “Inference and predictions” → “Inference or predictions”

L81: remove “prohibitive” as it does not add anything to this sentence.

L82-85: you might want to draw comparison to approximate Bayesian methods for fitting JSDMs instead of MLE here, such as Inla or empirical Bayes methods.

L88-91: This is an excellent sentence!

L93-94: “...explore what extent it can be … hardware.” → “explore to what extent it can be enhanced, when substituting the original model fitting implementation with an algorithm suited for HPC hardware” or similar. Many of the sentences in the manuscript can be made more succinct without losing information, making the whole manuscript more enjoyable to read.

L96: This sentence makes me wonder if I can use your new implementation for parallel computing with CPU instead, can the authors add something on that here? (I later read a statement on this on L143, but the point remains that the statement should be moved up in the text)

L101: remove “numerous” since this really depends on the machine/cluster that is being utilized

L102: “decomposes” is perhaps not the right word here. A more appropriate phrasing might be “Specifically, the bulk of operations within the block-Gibbs algorithm consists are algebraic in nature, so that they can be formulated within…”

L103: “These computations can be parallelized” → “The computations can then be parallelized”

L104: remove “multiple” as it is already implied. Also remove “the” before “simultaneous”

L107-108: rephrase to something like “We redesigned and re-implemented the Hmsc-R block-Gibbs sampler routine in the Python language using the …”

L127: “with TensorFlow-based” → “with the TensorFlow-based”

L154: remove “numerous” again

L188: “experimental design” → “design”. The experimental is implied by the first “experiments”

L199-201: this seems a bit odd, why do the authors not just introduce a second example dataset that actually has a Phylogeny available with it?

L224-237: I would be happy for further clarification here. Although (apparently) parallelized GPU computation is not possible for the models that use a NNGP, parallel CPU computations are still available? No speed-up compared to Hmsc routines implemented in R seems very surprising, especially as e.g., Finley et al. (2019) conclude that NNGP approximations are highly parallelizable.

References

Finley, Andrew O., et al. "Efficient algorithms for Bayesian nearest neighbor Gaussian processes." Journal of Computational and Graphical Statistics 28.2 (2019): 401-414.

Hui, Francis KC, et al. "Spatiotemporal joint species distribution modelling: A basis function approach." Methods in Ecology and Evolution 14.8 (2023): 2150-2164.

Kidzinski, L., Hui, F. K., Warton, D. I., & Hastie, T. J. (2022). Generalized Matrix Factorization: efficient algorithms for fitting generalized linear latent variable models to large data arrays. Journal of Machine Learning Research, 23(291), 1-29.

Niku, J., Brooks, W., Herliansyah, R., Hui, F. K. C., Korhonen, P., Taskinen, S., van der Veen, B., and Warton, D. I. (2024). gllvm: Generalized Linear Latent Variable Models. R package version 1.4.5.

Popovic, Gordana C., Francis KC Hui, and David I. Warton. "Fast model‐based ordination with copulas." Methods in Ecology and Evolution 13.1 (2022): 194-202.

**Have the authors made all data and (if applicable) computational code underlying the findings in their manuscript fully available?**

Reviewer #1: Yes

Reviewer #2: Yes

Reviewer #3: Yes

PLOS authors have the option to publish the peer review history of their article (what does this mean?). If published, this will include your full peer review and any attached files.

Reviewer #1: No

Reviewer #2: No

Reviewer #3: No
---

## [Decision Letter · Decision Letter 1]

26 Jul 2024

Dear Dr Rahman,

We are pleased to inform you that your manuscript 'Accelerating joint species distribution modelling with Hmsc-HPC by GPU porting' has been provisionally accepted for publication in PLOS Computational Biology.

Best regards,

Samuel V. Scarpino

Academic Editor

PLOS Computational Biology

James O'Dwyer

Section Editor

PLOS Computational Biology

Reviewer's Responses to Questions

**Comments to the Authors:**

Reviewer #1: The authors did an outstanding job responding to my comments and concerns. I appreciate the work the authors put into the revisions, and have no further comments. This will be an important contribution to the JSDM and statistical software community in the ecological sciences.

Reviewer #3: I thank the authors for a thorough revision, and have not further comments to improve their manuscript.

Reviewer #4: This is a review of the manuscript entitled "Accelerating joint species distribution modelling with Hmsc-HPC by GPU porting" which has been re-submitted for review after undergoing revisions. I'll note that this is my first review of the manuscript, I was not involved in previous review rounds. I'm basing this review off of the revised submission and response to reviewers only, I did not read the marked-up text changes from the original submission.

I think the authors did a commendable job responding to the original round of reviews. I support the author's decision to push back against conducting a large-scale comparison (having done similar levels of comparisons myself). While it would be a very useful study to conduct I feel it is beyond the scope of this paper presenting a new software implementation, would be able to stand alone as a publication of it's own, and that mixing the two would most likely result in the comparison drowning out the implementation in the paper.

Aside from a few minor edit suggestions to the figures below I have no further recommendations for the manuscript itself.

- Fig 1: The model name in 2a here is HMSC-R which is different to what you use in the manuscript: HMSC for the modelling framework and Hmsc for the software (or Hmsc R package). The hyphen notation was only used for Hmsc-HPC

- Fig 1: In the right-most red box "edges" should be capitalised

- Fig 2: A stylistic quibble that the authors can ignore if they wish, but I vastly prefer plot/axis/legend labels to start with a capital letter, and for the plot IDs to be in "A)" format. This improves readability.

I sign all reviews that aren't double blind.

David Wilkinson

The University of Melbourne

**Have the authors made all data and (if applicable) computational code underlying the findings in their manuscript fully available?**

Reviewer #1: Yes

Reviewer #3: Yes

Reviewer #4: Yes

PLOS authors have the option to publish the peer review history of their article (what does this mean?). If published, this will include your full peer review and any attached files.

Reviewer #1: No

Reviewer #3: No

Reviewer #4: **Yes: **David Wilkinson

---

## [Editor Report · Acceptance letter]

26 Aug 2024

PCOMPBIOL-D-24-00267R1 

Accelerating joint species distribution modelling with Hmsc-HPC by GPU porting

Dear Dr Rahman,

I am pleased to inform you that your manuscript has been formally accepted for publication in PLOS Computational Biology. Your manuscript is now with our production department and you will be notified of the publication date in due course.

With kind regards,

Anita Estes
